# Disease Trajectory Maps

**Peter Schulam**
Dept. of Computer Science
Johns Hopkins University
Baltimore, MD 21218
pschulam@cs.jhu.edu

**Raman Arora**
Dept. of Computer Science
Johns Hopkins University
Baltimore, MD 21218
arora@cs.jhu.edu

## Abstract

Medical researchers are coming to appreciate that many diseases are in fact complex, heterogeneous syndromes composed of subpopulations that express different variants of a related complication. Longitudinal data extracted from individual electronic health records (EHR) offer an exciting new way to study subtle differences in the way these diseases progress over time. In this paper, we focus on answering two questions that can be asked using these databases of longitudinal EHR data. First, we want to understand whether there are individuals with similar *disease trajectories* and whether there are a small number of *degrees of freedom* that account for differences in trajectories across the population. Second, we want to understand how important clinical outcomes are associated with disease trajectories. To answer these questions, we propose the Disease Trajectory Map (DTM), a novel probabilistic model that learns low-dimensional representations of sparse and irregularly sampled longitudinal data. We propose a stochastic variational inference algorithm for learning the DTM that allows the model to scale to large modern medical datasets. To demonstrate the DTM, we analyze data collected on patients with the complex autoimmune disease, scleroderma. We find that DTM learns meaningful representations of disease trajectories and that the representations are significantly associated with important clinical outcomes.

## 1 Introduction

Longitudinal data is becoming increasingly important in medical research and practice. This is due, in part, to the growing adoption of electronic health records (EHRs), which capture snapshots of an individual's state over time. These snapshots include clinical observations (apparent symptoms and vital sign measurements), laboratory test results, and treatment information. In parallel, medical researchers are beginning to recognize and appreciate that many diseases are in fact complex, highly heterogeneous syndromes [Craig, 2008] and that individuals may belong to disease subpopulations or *subtypes* that express similar sets of symptoms over time (see e.g. Saria and Goldenberg [2015]). Examples of such diseases include asthma [Lötvall et al., 2011], autism [Wiggins et al., 2012], and COPD [Castaldi et al., 2014]. The data captured in EHRs can help better understand these complex diseases. EHRs contain many types of observations and the ability to track their progression can help bring in to focus the subtle differences across individual disease expression.

In this paper, we focus on two exploratory questions that we can begin to answer using longitudinal EHR data. First, we want to discover whether there are individuals with similar *disease trajectories* and whether there are a small number of *degrees of freedom* that account for differences across a heterogeneous population. A low-dimensional characterization of trajectories and how they differ can yield insights into the biological underpinnings of the disease. In turn, this may motivate new targeted therapies. In the clinic, physicians can analyze an individual's clinical history to estimate the low-dimensional representation of the trajectory and can use this knowledge to make more accurate prognoses and guide treatment decisions by comparing against representations of past trajectories. Second, we would like to know whether individuals with similar clinical outcomes (e.g. death, severe organ damage, or development of comorbidities) have similar disease trajectories. In complex diseases, individuals are often at risk of developing a number of severe complications and

clinicians rarely have access to accurate prognostic biomarkers. Discovering associations between target outcomes and trajectory patterns may both generate new hypotheses regarding the causes of these outcomes and help clinicians to better anticipate the event using an individual's clinical history.

**Contributions.** Our approach to simultaneously answering these questions is to embed individual disease trajectories into a low-dimensional vector space wherein similarity in the embedded space implies that two individuals have similar trajectories. Such a representation would naturally answer our first question, and could also be used to answer the second by comparing distributions over representations across groups defined by different outcomes. To learn these representations, we introduce a novel probabilistic model of longitudinal data, which we term the Disease Trajectory Map (DTM). In particular, the DTM models the trajectory over time of a single *clinical marker*, which is an observation or measurement recorded over time by clinicians that is used to track the progression of a disease (see e.g. Schulam et al. [2015]). Examples of clinical markers are pulmonary function tests or creatinine laboratory test results, which track lung and kidney function respectively. The DTM discovers low-dimensional (e.g. 2D or 3D) latent representations of clinical marker trajectories that are easy to visualize. We describe a stochastic variational inference algorithm for estimating the posterior distribution over the parameters and individual-specific representations, which allows our model to be easily applied to large datasets. To demonstrate the DTM, we analyze clinical marker data collected on individuals with the complex autoimmune disease scleroderma (see e.g. Allanore et al. [2015]). We find that the learned representations capture interesting subpopulations consistent with previous findings, and that the representations suggest associations with important clinical outcomes not captured by alternative representations.

## 1.1 Background and Related Work

Clinical marker data extracted from EHRs is a by-product of an individual's interactions with the healthcare system. As a result, the time series are often irregularly sampled (the time between samples varies within and across individuals), and may be extremely sparse (it is not unusual to have a single observation for an individual). To aid the following discussion, we briefly introduce notation for this type of data. We use $m$ to denote the number of individual disease trajectories recorded in a given dataset. For each individual, we use $n_i$ to denote the number of observations. We collect the observation times for subject $i$ into a column vector $\mathbf{t}_i \triangleq [t_{i1}, \ldots, t_{in_i}]^\top$ (sorted in non-decreasing order) and the corresponding measurements into a column vector $\mathbf{y}_i \triangleq [y_{i1}, \ldots, y_{in_i}]^\top$. Our goal is to embed the pair $(\mathbf{t}_i, \mathbf{y}_i)$ into a low-dimensional vector space wherein similarity between two embeddings $\mathbf{x}_i$ and $\mathbf{x}_j$) implies that the trajectories have similar shapes. This is commonly done using *basis representations* of the trajectories.

**Fixed basis representations.** In the statistics literature, $(\mathbf{t}_i, \mathbf{y}_i)$ is often referred to as *unbalanced longitudinal data*, and is commonly analyzed using linear mixed models (LMMs) [Verbeke and Molenberghs, 2009]. In their simplest form, LMMs assume the following probabilistic model:

$$\mathbf{w}_i \mid \Sigma \sim \mathcal{N}(0, \Sigma) \ , \ \mathbf{y}_i \mid \mathrm{B}_i, \mathbf{w}_i, \mu, \sigma^2 \sim \mathcal{N}(\mu + \mathrm{B}_i \mathbf{w}_i, \sigma^2 \mathrm{I}_{n_i}). \tag{1}$$

The matrix $\mathrm{B}_i \in \mathbb{R}^{n_i \times d}$ is known as the *design matrix*, and can be used to capture non-linear relationships between the observation times $\mathbf{t}_i$ and measurements $\mathbf{y}_i$. Its rows are comprised of $d$-dimensional basis expansions of each observation time $\mathrm{B}_i = [\mathbf{b}(t_{i1}), \cdots, \mathbf{b}(t_{in_i})]^\top$. Common choices of $\mathbf{b}(\cdot)$ include polynomials, splines, wavelets, and Fourier series. The particular basis used is often carefully crafted by the analyst depending on the nature of the trajectories and on the desired structure (e.g. invariance to translations and scaling) in the representation [Brillinger, 2001]. The design matrix can therefore make the LMM remarkably flexible despite its simple parametric probabilistic assumptions. Moreover, the prior over $\mathbf{w}_i$ and the conjugate likelihood make it straightforward to fit $\mu$, $\Sigma$, and $\sigma^2$ using EM or Bayesian posterior inference.

After estimating the model parameters, we can estimate the coefficients $\mathbf{w}_i$ of a given clinical marker trajectory using the posterior distribution, which embeds the trajectory in a Euclidean space. To flexibly capture complex trajectory shapes, however, the basis must be high-dimensional, which makes interpretability of the representations challenging. We can use low-dimensional summaries such as the projection on to a principal subspace, but these are not necessarily substantively meaningful. Indeed, much research has gone into developing principal direction post-processing techniques (e.g. Kaiser [1958]) or alternative estimators that enhance interpretability (e.g. Carvalho et al. [2012]).

**Data-adaptive basis representations.** A set of related, but more flexible, techniques comes from functional data analysis where observations are functions (i.e. trajectories) assumed to be sampled

from a stochastic process and the goal is to find a parsimonious representation for the data [Ramsay et al., 2002]. Functional principal component analysis (FPCA), one of the most popular techniques in functional data analysis, expresses functional data in the orthonormal basis given by the eigenfunctions of the auto-covariance operator. This representation is optimal in the sense that no other representation captures more variation [Ramsay, 2006]. The idea itself can be traced back to early independent work by Karhunen and Loeve and is also referred to as the Karhunen-Loeve expansion [Watanabe, 1965]. While numerous variants of FPCA have been proposed, the one that is most relevant to the problem at hand is that of sparse FPCA [Castro et al., 1986, Rice and Wu, 2001] where we allow sparse, irregularly sampled data as is common in longitudinal data analysis. To deal with the sparsity, Rice and Wu [2001] used LMMs to model the auto-covariance operator. In very sparse settings, however, LMMs can suffer from numerical instability of covariance matrices in high dimensions. James et al. [2000] addressed this by constraining the rank of the covariance matrices—we will refer to this model as the reduced-rank LMM, but note that it is a variant of sparse FPCA. Although sparse FPCA represents trajectories using a data-driven basis, the basis is restricted to lie in a linear subspace of a fixed basis, which may be overly restrictive. Other approaches to learning a functional basis include Bayesian estimation of B-spline parameters (e.g. [Bigelow and Dunson, 2012]) and placing priors over reproducing kernel Hilbert spaces (e.g. [MacLehose and Dunson, 2009]). Although flexible, these two approaches do not learn a low-dimensional representation.

**Cluster-based representations.** Mixture models and clustering approaches are also commonly used to represent and discover structure in time series data. Marlin et al. [2012] cluster time series data from the intensive care unit (ICU) using a mixture model and use cluster membership to predict outcomes. Schulam and Saria [2015] describe a probabilistic model that represents trajectories using a hierarchy of features, which includes "subtype" or cluster membership. LMMs have also been extended to have nonparametric Dirichlet process priors over the coefficients (e.g. Kleinman and Ibrahim [1998]), which implicitly induce clusters in the data. Although these approaches flexibly model trajectory data, the structure they recover is a partition, which does not allow us to compare all trajectories in a coherent way as we can in a vector space.

**Lexicon-based representations.** Another line of research has investigated the discovery of motifs or repeated patterns in continuous time-series data for the purposes of succinctly representing the data as a string of elements of the discovered lexicon. These include efforts in the speech processing community to identify sub-word units (parts of words comparable to phonemes) in a data-driven manner [Varadarajan et al., 2008, Levin et al., 2013]. In computational healthcare, Saria et al. [2011] propose a method for discovering deformable motifs that are repeated in continuous time series data. These methods are, in spirit, similar to discretization approaches such as symbolic aggregate approximation (SAX) [Lin et al., 2007] and piecewise aggregate approximation (PAA) [Keogh et al., 2001] that are popular in data mining, and aim to find compact descriptions of sequential data, primarily for the purposes of indexing, search, anomaly detection, and information retrieval. The focus in this paper is to learn representations for entire trajectories rather than discover a lexicon. Furthermore, we focus on learning a representation in a vector space where similarities among trajectories are captured through the standard inner product on $\mathbb{R}^d$.

## 2 Disease Trajectory Maps

To motivate Disease Trajectory Maps (DTM), we begin with the reduced-rank LMM proposed by James et al. [2000]. We show that the reduced-rank LMM defines a Gaussian process with a covariance function that linearly depends on trajectory-specific representations. To define DTMs, we then use the kernel trick to make the dependence non-linear. Let $\mu \in \mathbb{R}$ be the marginal mean of the observations, $\mathrm{F} \in \mathbb{R}^{d \times q}$ be a rank-$q$ matrix, and $\sigma^2$ be the variance of measurement errors. As a reminder, $\mathbf{y}_i \in \mathbb{R}^{n_i}$ denotes the vector of observed trajectory measurements, $\mathrm{B}_i \in \mathbb{R}^{n_i \times d}$ denotes the individual's design matrix, and $\mathbf{x}_i \in \mathbb{R}^q$ denotes the individual's representation. James et al. [2000] define the reduced-rank LMM using the following conditional distribution:

$$\mathbf{y}_i \mid \mathrm{B}_i, \mathbf{x}_i, \mu, \mathrm{F}, \sigma^2 \sim \mathcal{N}(\mu + \mathrm{B}_i \mathrm{F} \mathbf{x}_i, \sigma^2 \mathrm{I}_{n_i}). \tag{2}$$

They assume an isotropic normal prior over $\mathbf{x}_i$ and marginalize to obtain the observed-data log-likelihood, which is then optimized with respect to $\{\mu, \mathrm{F}, \sigma^2\}$. As in Lawrence [2004], we instead optimize $\mathbf{x}_i$ and marginalize $\mathrm{F}$. By assuming a normal prior $\mathcal{N}(\mathbf{0}, \alpha \mathrm{I}_q)$ over the rows of $\mathrm{F}$ and marginalizing we obtain:

$$\mathbf{y}_i \mid \mathrm{B}_i, \mathbf{x}_i, \mu, \sigma^2, \alpha \sim \mathcal{N}(\mu, \alpha \langle \mathbf{x}_i, \mathbf{x}_i \rangle \mathrm{B}_i \mathrm{B}_i^\top + \sigma^2 \mathrm{I}_{n_i}). \tag{3}$$

Note that by marginalizing over F, we induce a joint distribution over all trajectories in the dataset. Moreover, this joint distribution is a Gaussian process with mean $\mu$ and the following covariance function defined across trajectories that depends on times $\{\mathbf{t}_i, \mathbf{t}_j\}$ and representations $\{\mathbf{x}_i, \mathbf{x}_j\}$:

$$\text{Cov}(\mathbf{y}_i, \mathbf{y}_j \mid \mathrm{B}_i, \mathrm{B}_j, \mathbf{x}_i, \mathbf{x}_j, \mu, \sigma^2, \alpha) = \alpha \langle \mathbf{x}_i, \mathbf{x}_j \rangle \mathrm{B}_i \mathrm{B}_j^\top + \mathbb{I}[i = j] \,(\sigma^2 \mathrm{I}_{n_i}). \tag{4}$$

This reformulation of the reduced-rank LMM highlights that the covariance across trajectories $i$ and $j$ depends on the inner product between the two representations $\mathbf{x}_i$ and $\mathbf{x}_j$, and suggests that we can non-linearize the dependency with an inner product in an expanded feature space using the "kernel trick". Let $k(\cdot, \cdot)$ denote a non-linear kernel defined over the representations with parameters $\boldsymbol{\theta}$, then we have:

$$\text{Cov}(\mathbf{y}_i, \mathbf{y}_j \mid \mathrm{B}_i, \mathrm{B}_j, \mathbf{x}_i, \mathbf{x}_j, \mu, \sigma^2, \boldsymbol{\theta}) = k(\mathbf{x}_i, \mathbf{x}_j) \mathrm{B}_i \mathrm{B}_j^\top + \mathbb{I}[i = j] \,(\sigma^2 \mathrm{I}_{n_i}). \tag{5}$$

Let $\mathbf{y} \triangleq [\mathbf{y}_1^\top, \ldots, \mathbf{y}_m^\top]^\top$ denote the column vector obtained by concatenating the measurement vectors from each trajectory. The joint distribution over $\mathbf{y}$ is a multivariate normal:

$$\mathbf{y} \mid \mathrm{B}_{1:m}, \mathbf{x}_{1:m}, \mu, \sigma^2, \boldsymbol{\theta} \sim \mathcal{N}(\mu, \Sigma_{\text{DTM}} + \sigma^2 \mathrm{I}_n), \tag{6}$$

where $\Sigma_{\text{DTM}}$ is a covariance matrix that depends on the times $\mathbf{t}_{1:m}$ (through design matrices $\mathrm{B}_{1:m}$) and representations $\mathbf{x}_{1:m}$. In particular, $\Sigma_{\text{DTM}}$ is a block-structured matrix with $m$ row blocks and $m$ column blocks. The block at the $i^{\text{th}}$ row and $j^{\text{th}}$ column is the covariance between $\mathbf{y}_i$ and $\mathbf{y}_j$ defined in (5). Finally, we place isotropic Gaussian priors over $\mathbf{x}_i$. We use Bayesian inference to obtain a posterior Gaussian process and to estimate the representations. We tune hyperparameters by maximizing the observed-data log likelihood. Note that our model is similar to the Bayesian GPLVM [Titsias and Lawrence, 2010], but models functional data instead of finite-dimensional vectors.

## 2.1 Learning and Inference in the DTM

As formulated, the model scales poorly to large datasets. Inference within each iteration of an optimization algorithm, for example, requires storing and inverting $\Sigma_{\text{DTM}}$, which requires $O(n^2)$ space and $O(n^3)$ time respectively, where $n \triangleq \sum_{i=1}^m n_i$ is the number of clinical marker observations. For modern datasets, where $n$ can be in the hundreds of thousands or millions, this is unacceptable. In this section, we approximate the log-likelihood using techniques from Hensman et al. [2013] that allow us to apply stochastic variational inference (SVI) [Hoffman et al., 2013].

**Inducing points.** Recent work in scaling Gaussian processes to large datasets has focused on the idea of *inducing points* [Snelson and Ghahramani, 2005, Titsias, 2009], which are a relatively small number of artificial observations of a Gaussian process that approximately capture the information contained in the training data. In general, let $\mathbf{f} \in \mathbb{R}^m$ denote observations of a GP at inputs $\{\mathbf{x}_i\}_{i=1}^m$ and $\mathbf{u} \in \mathbb{R}^p$ denote inducing points at inputs $\{\mathbf{z}_i\}_{i=1}^p$. Titsias [2009] constructs the inducing points as variational parameters by introducing an augmented probability model:

$$\mathbf{u} \sim \mathcal{N}(\mathbf{0}, \mathrm{K}_{pp}) \;, \;\; \mathbf{f} \mid \mathbf{u} \sim \mathcal{N}(\mathrm{K}_{mp}\mathrm{K}_{pp}^{-1}\mathbf{u}, \tilde{\mathrm{K}}_{mm}), \tag{7}$$

where $\mathrm{K}_{pp}$ is the Gram matrix between inducing points, $\mathrm{K}_{mm}$ is the Gram matrix between observations, $\mathrm{K}_{mp}$ is the cross Gram matrix between observations and inducing points, and $\tilde{\mathrm{K}}_{mm} \triangleq \mathrm{K}_{mm} - \mathrm{K}_{mp}\mathrm{K}_{pp}^{-1}\mathrm{K}_{pm}$. We can marginalize over $\mathbf{u}$ to construct a low-rank approximate covariance matrix, which is computationally cheaper to invert using the Woodbury identity. Alternatively, Hensman et al. [2013] extends these ideas by explicitly maintaining a variational distribution over $\mathbf{u}$ that d-separates the observations and satisfies the conditions required to apply SVI [Hoffman et al., 2013]. Let $\mathbf{y}_f = \mathbf{f} + \boldsymbol{\epsilon}$ where $\boldsymbol{\epsilon}$ is iid Gaussian noise with variance $\sigma^2$, then we use the following inequality to lower bound our data log-likelihood:

$$\log p(\mathbf{y}_f \mid \mathbf{u}) \geq \sum_{i=1}^m \mathbb{E}_{p(f_i \mid \mathbf{u})}[\log p(y_{fi} \mid f_i)]. \tag{8}$$

In the interest of space, we refer the interested reader to Hensman et al. [2013] for details.

**DTM evidence lower bound.** When marginalizing over the rows of F, we induced a Gaussian process over the trajectories, but by doing so we also implicitly induced a Gaussian process over the individual-specific basis coefficients. Let $\mathbf{w}_i \triangleq \mathrm{F}\mathbf{x}_i \in \mathbb{R}^d$ denote the basis weights implied by the mapping F and representation $\mathbf{x}_i$ in the reduced-rank LMM, and let $\mathbf{w}_{:,k}$ for $k \in [d]$ denote the $k^{\text{th}}$ coefficient of all individuals in the dataset. After marginalizing the $k^{\text{th}}$ row of F and applying the kernel trick, we see that the vector of coefficients $\mathbf{w}_{:,k}$ has a Gaussian process distribution

with mean zero and covariance function: $\mathrm{Cov}(w_{ik}, w_{jk}) = \alpha k(\mathbf{x}_i, \mathbf{x}_j)$. Moreover, the Gaussian processes across coefficients are statistically independent of one another. To lower bound the DTM log-likelihood, we introduce $p$ inducing points $\mathbf{u}_k$ for each vector of coefficients $\mathbf{w}_{:,k}$ with shared inducing point inputs $\{\mathbf{z}_i\}_{i=1}^p$. To refer to all inducing points simultaneously, we use $\mathrm{U} \triangleq [\mathbf{u}_1, \ldots, \mathbf{u}_d]$ and $\mathbf{u}$ to denote the "vectorized" form of $\mathrm{U}$ obtained by stacking its columns. Applying (8) we have:

$$\log p(\mathbf{y} \mid \mathbf{u}, \mathbf{x}_{1:m}) \geq \sum_{i=1}^m \mathbb{E}_{p(\mathbf{w}_i \mid \mathbf{u}, \mathbf{x}_i)}[\log p(\mathbf{y}_i \mid \mathbf{w}_i)]$$

$$= \sum_{i=1}^m \log \mathcal{N}(\mathbf{y}_i \mid \mu + \mathrm{B}_i \mathrm{U}^\top \mathrm{K}_{pp}^{-1} \mathbf{k}_i, \sigma^2 \mathrm{I}_{n_i}) - \frac{\tilde{k}_{ii}}{2\sigma^2} \mathrm{Tr}[\mathrm{B}_i^\top \mathrm{B}_i] \triangleq \sum_{i=1}^m \log \tilde{p}(\mathbf{y}_i \mid \mathbf{u}, \mathbf{x}_i), \quad (9)$$

where $\mathbf{k}_i \triangleq [k(\mathbf{x}_i, \mathbf{z}_1), \ldots, k(\mathbf{x}_i, \mathbf{z}_p)]^\top$ and $\tilde{k}_{ii}$ is the $i^{\text{th}}$ diagonal element of $\tilde{\mathrm{K}}_{mm}$. We can then construct the variational lower bound on $\log p(\mathbf{y})$:

$$\log p(\mathbf{y}) \geq \mathbb{E}_{q(\mathbf{u}, \mathbf{x}_{1:m})}[\log p(\mathbf{y} \mid \mathbf{u}, \mathbf{x}_{1:m})] - \mathrm{KL}(q(\mathbf{u}, \mathbf{x}_{1:m}) \| p(\mathbf{u}, \mathbf{x}_{1:m})) \quad (10)$$

$$\geq \sum_{i=1}^m \mathbb{E}_{q(\mathbf{u}, \mathbf{x}_i)}[\log \tilde{p}(\mathbf{y}_i \mid \mathbf{u}, \mathbf{x}_i)] - \mathrm{KL}(q(\mathbf{u}, \mathbf{x}_{1:m}) \| p(\mathbf{u}, \mathbf{x}_{1:m})), \quad (11)$$

where we use the lower bound in (9). Finally, to make the lower bound concrete we specify the variational distribution $q(\mathbf{u}, \mathbf{x}_{1:m})$ to be a product of independent multivariate normal distributions:

$$q(\mathbf{u}, \mathbf{x}_{1:m}) \triangleq \mathcal{N}(\mathbf{u} \mid \mathbf{m}, \mathrm{S}) \prod_{i=1}^m \mathcal{N}(\mathbf{x}_i \mid \mathbf{m}_i, \mathrm{S}_i), \quad (12)$$

where the variational parameters to be fit are $\mathbf{m}$, $\mathrm{S}$, and $\{\mathbf{m}_i, \mathrm{S}_i\}_{i=1}^m$.

**Stochastic optimization of the lower bound.** To apply SVI, we must be able to compute the gradient of the expected value of $\log \tilde{p}(\mathbf{y}_i \mid \mathbf{u}, \mathbf{x}_i)$ under the variational distributions. Because $\mathbf{u}$ and $\mathbf{x}_i$ are assumed to be independent in the variational posteriors, we can analyze the expectation in either order. Fix $\mathbf{x}_i$, then we see that $\log \tilde{p}(\mathbf{y}_i \mid \mathbf{u}, \mathbf{x}_i)$ depends on $\mathbf{u}$ only through the mean of the Gaussian density, which is a quadratic term in the log likelihood. Because $q(\mathbf{u})$ is multivariate normal, we can compute the expectation in closed form.

$$\mathbb{E}_{q(\mathbf{u})}[\log \tilde{p}(\mathbf{y}_i \mid \mathbf{u}, \mathbf{x}_i)] = \mathbb{E}_{q(\mathrm{U})}[\log \mathcal{N}(\mathbf{y}_i \mid \mu + (\mathrm{B}_i \otimes \mathbf{k}_i^\top \mathrm{K}_{pp}^{-1})\mathbf{u}, \sigma^2 \mathrm{I}_{n_i})] - \frac{\tilde{k}_{ii}}{2\sigma^2} \mathrm{Tr}[\mathrm{B}_i^\top \mathrm{B}_i]$$

$$= \log \mathcal{N}(\mathbf{y}_i \mid \mu + \mathrm{C}_i \mathbf{m}, \sigma^2 \mathrm{I}_{n_i})] - \frac{1}{2\sigma^2} \mathrm{Tr}[\mathrm{S}\mathrm{C}_i^\top \mathrm{C}_i] - \frac{\tilde{k}_{ii}}{2\sigma^2} \mathrm{Tr}[\mathrm{B}_i^\top \mathrm{B}_i],$$

where we have defined $\mathrm{C}_i \triangleq (\mathrm{B}_i \otimes \mathbf{k}_i^\top \mathrm{K}_{pp}^{-1})$ to be the *extended design matrix* and $\otimes$ is the Kronecker product. We now need to compute the expectation of this expression with respect to $q(\mathbf{x}_i)$, which entails computing the expectations of $\mathbf{k}_i$ (a vector) and $\mathbf{k}_i \mathbf{k}_i^\top$ (a matrix). In this paper, we assume an RBF kernel, and so the elements of the vector and matrix are all exponentiated quadratic functions of $\mathbf{x}_i$. This makes the expectations straightforward to compute given that $q(\mathbf{x}_i)$ is multivariate normal.[1] We therefore see that the expected value of $\log \tilde{p}(\mathbf{y}_i)$ can be computed in closed form under the assumed variational distribution.

We use the standard SVI algorithm to optimize the lower bound. We subsample the data, optimize the likelihood of each example in the batch with respect to the variational parameters over the representation $(\mathbf{m}_i, \mathrm{S}_i)$, and compute approximate gradients of the global variational parameters $(\mathbf{m}, \mathrm{S})$ and the hyperparameters. The likelihood term is conjugate to the prior over $\mathbf{u}$, and so we can compute the natural gradients with respect to the global variational parameters $\mathbf{m}$ and $\mathrm{S}$ [Hoffman et al., 2013, Hensman et al., 2013]. Additional details on the approximate objective and the gradients required for SVI are given in the supplement. We provide details on initialization, minibatch selection, and learning rates for our experiments in Section 3.

**Inference on new trajectories.** The variational distribution over the inducing point values $\mathbf{u}$ can be used to approximate a *posterior process* over the basis coefficients $\mathbf{w}_i$ [Hensman et al., 2013]. Therefore, given a representation $\mathbf{x}_i$, we have that

$$w_{ik} \mid \mathbf{x}_i, \mathbf{m}, \mathbf{S} \sim \mathcal{N}(\mathbf{k}_i^\top \mathrm{K}_{pp}^{-1} \mathbf{m}_k, \tilde{k}_{ii} + \mathbf{k}_i^\top \mathrm{K}_{pp}^{-1} \mathrm{S}_{kk} \mathrm{K}_{pp}^{-1} \mathbf{k}_i), \quad (13)$$

where $\mathbf{m}_k$ is the approximate posterior mean of the $k^{\text{th}}$ column of U and $S_{kk}$ is its covariance. The approximate joint posterior distribution over all coefficients can be shown to be multivariate normal. Let $\boldsymbol{\mu}(\mathbf{x}_i)$ be the mean of this distribution given representation $\mathbf{x}_i$ and $\Sigma(\mathbf{x}_i)$ be the covariance, then the posterior predictive distribution over a new trajectory $\mathbf{y}_*$ given the representation $\mathbf{x}_*$ is

$$\mathbf{y}_* \mid \mathbf{x}_* \sim \mathcal{N}(\mu + \mathrm{B}_*\boldsymbol{\mu}(\mathbf{x}_*), \mathrm{B}_*\Sigma(\mathbf{x}_*)\mathrm{B}_*^\top + \sigma^2 \mathrm{I}_{n_*}). \qquad (14)$$

We can then approximately marginalize with respect to the prior over $\mathbf{x}_*$ or a variational approximation of the posterior given a partial trajectory using a Monte Carlo estimate.

## 3    Experiments

We now use DTM to analyze clinical marker trajectories of individuals with the autoimmune disease, scleroderma [Allanore et al., 2015]. Scleroderma is a heterogeneous and complex chronic autoimmune disease. It can potentially affect many of the visceral organs, such as the heart, lungs, kidneys, and vasculature. Any given individual may experience only a subset of complications, and the timing of the symptoms relative to disease onset can vary considerably across individuals. Moreover, there are no known biomarkers that accurately predict an individual's disease course. Clinicians and medical researchers are therefore interested in characterizing and understanding disease progression patterns. Moreover, there are a number of clinical outcomes responsible for the majority of morbidity among patients with scleroderma. These include congestive heart failure, pulmonary hypertension and pulmonary arterial hypertension, gastrointestinal complications, and myositis [Varga et al., 2012]. We use the DTM to study associations between these outcomes and disease trajectories.

We study two scleroderma clinical markers. The first is the percent of predicted forced vital capacity (PFVC); a pulmonary function test result measuring lung function. PFVC is recorded in percentage points, and a higher value (near 100) indicates that the individual's lungs are functioning as expected. The second clinical marker that we study is the total modified Rodnan skin score (TSS). Scleroderma is named after its effect on the skin, which becomes hard and fibrous during periods of high disease activity. Because it is the most clinically apparent symptom, many of the current sub-categorizations of scleroderma depend on an individual's pattern of skin disease activity over time [Varga et al., 2012]. To systematically monitor skin disease activity, clinicians use the TSS which is a quantitative score between 0 and 55 computed by evaluating skin thickness at 17 sites across the body (higher scores indicate more active skin disease).

### 3.1    Experimental Setup

For our experiments, we extract trajectories from the Johns Hopkins Hospital Scleroderma Center's patient registry; one of the largest in the world. For both PFVC and TSS, we study the trajectory from the time of first symptom until ten years of follow-up. The PFVC dataset contains trajectories for 2,323 individuals and the TSS dataset contains 2,239 individuals. The median number of observations per individuals is 3 for the PFVC data and 2 for the TSS data. The maximum number of observations is 55 and 22 for PFVC and TSS respectively.

We present two sets of results. First, we visualize groups of similar trajectories obtained by clustering the representations learned by DTM. Although not quantitative, we use these visualizations as a way to check that the DTM uncovers subpopulations that are consistent with what is currently known about scleroderma. Second, we use the learned representations of trajectories obtained using the LMM, the reduced-rank LMM (which we refer to as FPCA), and the DTM to statistically test for relationships between important clinical outcomes and learned disease trajectory representations.

For all experiments and all models, we use a common 5-dimensional B-spline basis composed of degree-2 polynomials (see e.g. Chapter 20 in Gelman et al. [2014]). We choose knots using the percentiles of observation times across the entire training set [Ramsay et al., 2002]. For LMM and FPCA, we use EM to fit model parameters. To fit the DTM, we use the LMM estimate to set the mean $\mu$, noise $\sigma^2$, and average the diagonal elements of $\Sigma$ to set the kernel scale $\alpha$. Length-scales $\ell$ are set to 1. For these experiments, we do not learn the kernel hyperparameters during optimization. We initialize the variational means over $\mathbf{x}_i$ using the first two unit-scaled principal components of $\mathbf{w}_i$ estimated by LMM and set the variational covariances to be diagonal with standard deviation 0.1. For both PFVC and TSS, we use minibatches of size 25 and learn for a total of five epochs (passes over the training data). The initial learning rate for $\mathbf{m}$ and S is 0.1 and decays as $t^{-1}$ for each epoch $t$.

### 3.2    Qualitative Analysis of Representations

The DTM returns approximate posteriors over the representations $\mathbf{x}_i$ for all individuals in the training set. We examine these posteriors for both the PFVC and TSS datasets to check for consistency with

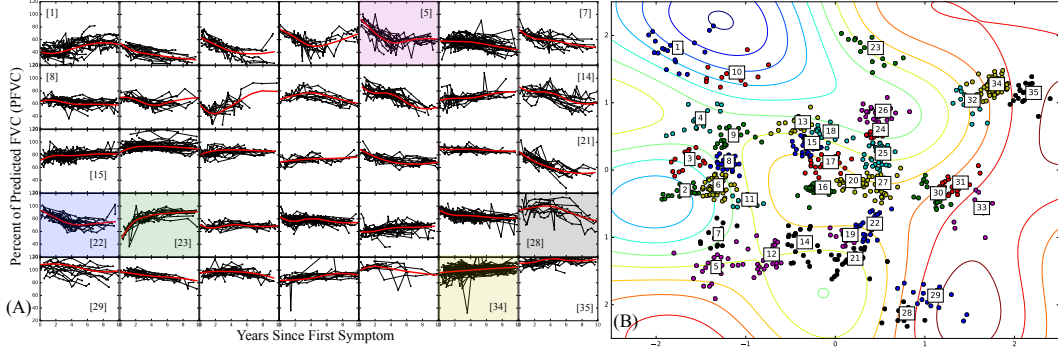

Figure 1: (A) Groups of PFVC trajectories obtained by hierarchical clustering of DTM representations. (B) Trajectory representations are color-coded and labeled according to groups shown in (A). Contours reflect posterior GP over the second B-spline coefficient (blue contours denote smaller values, red denote larger values).

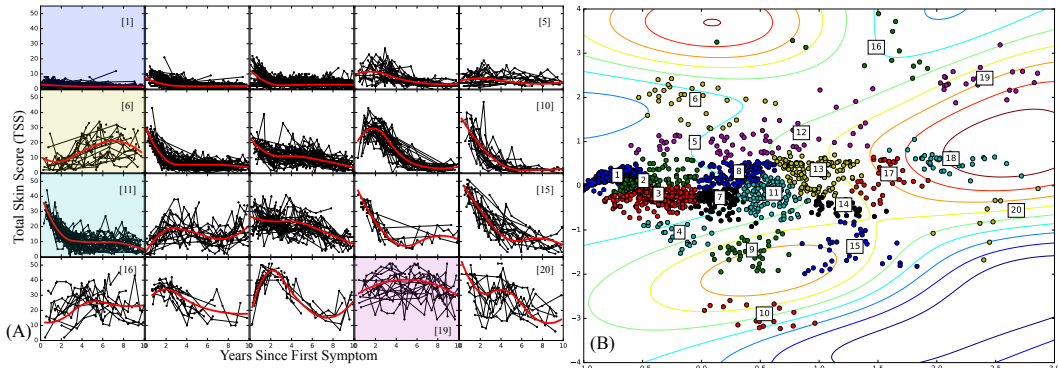

Figure 2: Same presentation as in Figure 1 but for TSS trajectories.

what is currently known about scleroderma disease trajectories. In Figure 1 (A) we show groups of trajectories uncovered by clustering the posterior means over the representations, which are plotted in Figure 1 (B). Many of the groups shown here align with other work on scleroderma lung disease subtypes (e.g. Schulam et al. [2015]). In particular, we see rapidly declining trajectories (group [5]), slowly declining trajectories (group [22]), recovering trajectories (group [23]), and stable trajectories (group [34]). Surprisingly, we also see a group of individuals who we describe as "late decliners" (group [28]). These individuals are stable for the first 5-6 years, but begin to decline thereafter. This is surprising because the onset of scleroderma-related lung disease is currently thought to occur early in the disease course [Varga et al., 2012]. In Figure 2 (A) we show clusters of TSS trajectories and the corresponding mean representations in Figure 2 (B). These trajectories corroborate what is currently known about skin disease in scleroderma. In particular, we see individuals who have minimal activity (e.g. group [1]) and individuals with early activity that later stabilizes (e.g. group [11]), which correspond to what are known as the limited and diffuse variants of scleroderma [Varga et al., 2012]. We also find that there are a number of individuals with increasing activity over time (group [6]) and some whose activity remains high over the ten year period (group [19]). These patterns are not currently considered to be canonical trajectories and warrant further investigation.

### 3.3 Associations between Representations and Clinical Outcomes

To quantitatively evaluate the low-dimensional representations learned by the DTM, we statistically test for relationships between the representations of clinical marker trajectories and important clinical outcomes. We compare the inferences of the hypothesis test with those made using representations derived from the LMM and FPCA baselines. For the LMM, we project $\mathbf{w}_i$ into its 2-dimensional principal subspace. For FPCA, we learn a rank-2 covariance, which learns 2-dimensional representations. To establish that the models are all equally expressive and achieve comparable generalization error, we present held-out data log-likelihoods in Table 1, which are estimated using 10-fold cross-validation. We see that the models are roughly equivalent with respect to generalization error.

To test associations between clinical outcomes and learned representations, we use a kernel density estimator test [Duong et al., 2012] to test the null hypothesis that the distributions across subgroups with and without the outcome are equivalent. The $p$-values obtained are listed in Table 2. As a point of

Table 1: Disease Trajectory Held-out Log-Likelihoods

| | PFVC | | TSS | |
| Model | Subj. LL | Obs. LL | Subj. LL | Obs. LL |
|---|---|---|---|---|
| LMM | -17.59 ($\pm$ 1.18) | -3.95 ($\pm$ 0.04) | -13.63 ($\pm$ 1.41) | -3.47 ($\pm$ 0.05) |
| FPCA | -17.89 ($\pm$ 1.19) | -4.03 ($\pm$ 0.02) | -13.76 ($\pm$ 1.42) | -3.47 ($\pm$ 0.05) |
| DTM | -17.74 ($\pm$ 1.23) | -3.98 ($\pm$ 0.03) | -13.25 ($\pm$ 1.38) | -3.32 ($\pm$ 0.06) |

Table 2: P-values under the null hypothesis that the distributions of trajectory representations are the same across individuals with and without clinical outcomes. Lower values indicate stronger support for rejection.

| | PFVC | | | TSS | | |
| Outcome | LMM | FPCA | DTM | LMM | FPCA | DTM |
|---|---|---|---|---|---|---|
| Congestive Heart Failure | 0.170 | 0.081 | 0.013 | 0.107 | 0.383 | 0.189 |
| Pulmonary Hypertension | 0.270 | *0.000 | *0.000 | 0.485 | 0.606 | 0.564 |
| Pulmonary Arterial Hypertension | 0.013 | 0.020 | *0.002 | 0.712 | 0.808 | 0.778 |
| Gastrointestinal Complications | 0.328 | 0.073 | 0.347 | 0.026 | 0.035 | 0.011 |
| Myositis | 0.337 | *0.002 | *0.004 | *0.000 | *0.002 | *0.000 |
| Interstitial Lung Disease | *0.000 | *0.000 | *0.000 | 0.553 | 0.515 | 0.495 |
| Ulcers and Gangrene | 0.410 | 0.714 | 0.514 | 0.573 | 0.316 | *0.009 |

reference, we include two clinical outcomes that should be clearly related to the two clinical markers. Interstitial lung disease is the most common cause of lung damage in scleroderma [Varga et al., 2012], and so we confirm that the null hypothesis is rejected for all three PFVC representations. Similarly, for TSS we expect ulcers and gangrene to be associated with severe skin disease. In this case, only the representations learned by DTM reveal this relationship. For the remaining outcomes, we see that FPCA and DTM reveal similar associations, but that only DTM suggests a relationship with pulmonary arterial hypertension (PAH). Presence of fibrosis (which drives lung disease progression) has been shown to be a risk factor in the development of PAH (see Chapter 36 of Varga et al. [2012]), but only the representations learned by DTM corroborate this association (see Figure 3).

## 4 Conclusion

We presented the Disease Trajectory Map (DTM), a novel probabilistic model that learns low-dimensional embeddings of sparse and irregularly sampled clinical time series data. The DTM is a reformulation of the LMM. We derived it using an approach comparable to that of Lawrence [2004] in deriving the Gaussian process latent variable model (GPLVM) from probabilistic principal component analysis (PPCA) [Tipping and Bishop, 1999], and indeed the DTM can be interpreted as a "twin kernel" GPLVM (briefly discussed in the concluding paragraphs) over functional observations. The DTM can also be viewed as an LMM with a "warped" Gaussian prior over the random effects (see e.g. Damianou et al. [2015] for a discussion of distributions induced by mapping Gaussian random variables through non-linear maps). We demonstrated the model by analyzing data extracted from one of the nation's largest scleroderma patient registries, and found that the DTM discovers structure among trajectories that is consistent with previous findings and also uncovers several surprising disease trajectory shapes. We also explore associations between important clinical outcomes and the DTM's representations and found statistically significant differences in representations between outcome-defined groups that were not uncovered by two sets of baseline representations.

**Acknowledgments.** PS is supported by an NSF Graduate Research Fellowship. RA is supported in part by NSF BIGDATA grant IIS-1546482.

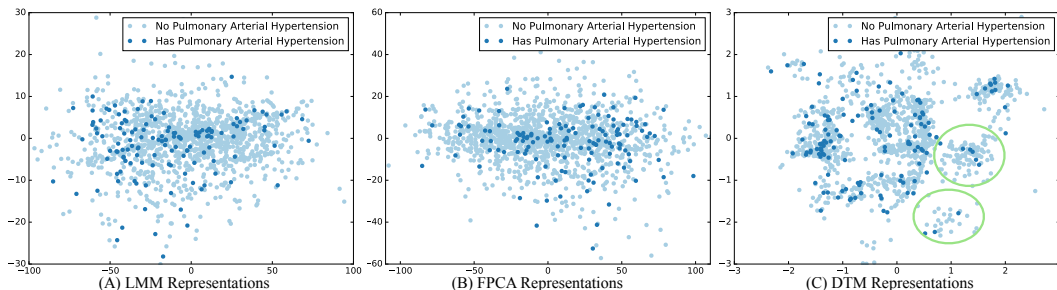

Figure 3: Scatter plots of PFVC representations for the three models color-coded by presence or absence of pulmonary arterial hypertension (PAH). Groups of trajectories with very few cases of PAH are circled in green.

## Footnotes

[1]Other kernels can be used instead, but the expectations may not have closed form expressions.

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
