[Supplementary Material]

# Disease Trajectory Maps: Supplementary Material

**Peter Schulam**
Dept. of Computer Science
Johns Hopkins University
Baltimore, MD 21218
pschulam@cs.jhu.edu

**Raman Arora**
Dept. of Computer Science
Johns Hopkins University
Baltimore, MD 21218
arora@cs.jhu.edu

## 1   Derivation of Evidence Lower Bound

When marginalizing over the rows of F, we induced a Gaussian process over the trajectories, but by doing so we implicitly induced a Gaussian process over the individual-specific basis coefficients. Let $\mathbf{w}_i \triangleq \mathrm{F}\mathbf{x}_i \in \mathbb{R}^d$ denote the basis weights implied by the mapping F and representation $\mathbf{x}_i$ in the reduced-rank LMM, and let $\mathbf{w}_{:,k}$ for $k \in [d]$ denote the $k^{\text{th}}$ coefficient of all individuals in the dataset. After marginalizing the $k^{\text{th}}$ row of F and applying the kernel trick, we see that the vector of coefficients $\mathbf{w}_{:,k}$ has a Gaussian process distribution with mean 0 and covariance

$$\mathrm{Cov}(w_{ik}, w_{jk}) = \alpha k(\mathbf{x}_i, \mathbf{x}_j). \tag{1}$$

Moreover, the Gaussian processes across coefficients are statistically independent of one another. To construct our approximate objective, we first approximate each of the $d$ coefficient Gaussian processes by introducing $p$ inducing points (see e.g. Snelson and Ghahramani [2005], Titsias [2009]) with values $\mathbf{u}_k \in \mathbb{R}^p$ for each $k \in [d]$ observed at common inputs $\mathbf{z}_i \in \mathbb{R}^q$ for $i \in [p]$. We assume that each $\mathbf{w}_{:,k}$ and $\mathbf{u}_k$ are sampled from a common Gaussian process, which implies the joint distribution:

$$\mathbf{u}_k \mid \boldsymbol{\theta} \sim \mathcal{N}(\mathbf{0}, \mathrm{K}_{pp}) \tag{2}$$

$$\mathbf{w}_k \mid \mathbf{u}_k, \boldsymbol{\theta} \sim \mathcal{N}(\mathrm{K}_{mp}\mathrm{K}_{pp}^{-1}\mathbf{u}_k, \tilde{\mathrm{K}}_{mm}). \tag{3}$$

where $\mathrm{K}_{pp}$ is the Gram matrix between inducing points, $\mathrm{K}_{mm}$ is the Gram matrix between individuals (based on their representations $\mathbf{x}_i$), $\mathrm{K}_{mp}$ is the cross Gram matrix between individuals and inducing points, and $\tilde{\mathrm{K}}_{mm} \triangleq \mathrm{K}_{mm} - \mathrm{K}_{mp}\mathrm{K}_{pp}^{-1}\mathrm{K}_{pm}$.

Now, we stack the inducing point values $\mathbf{u}_{1:d}$ into the columns of a matrix $\mathrm{U} \triangleq [\mathbf{u}_1, \ldots, \mathbf{u}_d]$. We will use $\mathbf{u}$ to denote the "vectorization" of U obtained by stacking the columns. Each row $i$ of U can be thought of as the vector of coefficients belonging to a single *inducing individual* which has an associated representation $\mathbf{z}_i \in \mathbb{R}^q$. Let $\mathbf{y} \triangleq [\mathbf{y}_1^\top, \ldots, \mathbf{y}_m^\top]^\top$ be the vector of concatenated trajectories and W be the matrix containing individual $i$'s coefficients $\mathbf{w}_i$ in each row, then following the derivation of Hensman et al. [2013], we can lower bound the conditional log-probability of $\mathbf{y}$ given $\mathbf{u}$ and $\mathbf{x}_{1:m}$:

$$\log p(\mathbf{y} \mid \mathbf{u}, \mathbf{x}_{1:m}) = \log \int p(\mathbf{y} \mid \mathrm{W}) p(\mathrm{W} \mid \mathbf{u}, \mathbf{x}_{1:m}) d\mathrm{W} \tag{4}$$

$$= \log \int \prod_{i=1}^{m} p(\mathbf{y}_i \mid \mathbf{w}_i) p(\mathrm{W} \mid \mathbf{u}, \mathbf{x}_{1:m}) d\mathrm{W} \tag{5}$$

$$\geq \int p(\mathrm{W} \mid \mathbf{u}, \mathbf{x}_{1:m}) \sum_{i=1}^{m} \log p(\mathbf{y}_i \mid \mathbf{w}_i) d\mathrm{W} \tag{6}$$

$$= \sum_{i=1}^{m} \mathbb{E}_{p(\mathbf{w}_i \mid \mathbf{u}, \mathbf{x}_i)}[\log p(\mathbf{y}_i \mid \mathbf{w}_i)]. \tag{7}$$

The expectation in each summand is easy to calculate because the mean of $\mathbf{y}_i$ is linearly dependent on $\mathbf{w}_i$ and because the conditional distribution $\mathbf{w}_i$ given $\mathbf{u}$ is multivariate normal. Specifically, we have that

$$\mathbf{w}_i \mid \mathbf{u}, \mathbf{x}_i \sim \mathcal{N}(\mathrm{U}^\top \mathrm{K}_{pp}^{-1} \mathbf{k}_i, \tilde{k}_{ii} \mathrm{I}_d), \tag{8}$$

where $\mathbf{k}_i$ is a column vector filled with the $i^{\text{th}}$ row of $\mathrm{K}_{mp}$ and $\tilde{k}_{ii}$ is the $i^{\text{th}}$ diagonal element of $\tilde{\mathrm{K}}_{mm}$. Together with the conditional distribution of $\mathbf{y}_i$ given $\mathbf{w}_i$, we have that each summand can be written as

$$\mathbb{E}_{p(\mathbf{w}_i \mid \mathbf{u}, \mathbf{x}_i)}[\log p(\mathbf{y}_i \mid \mathbf{w}_i)] \tag{9}$$

$$= -\frac{n_i}{2} \log 2\pi\sigma^2 - \frac{1}{2\sigma^2} \mathbb{E}_{p(\mathbf{w}_i \mid \mathbf{u}, \mathbf{x}_i)}[(\mathbf{y}_i - \mu - \mathrm{B}_i \mathbf{w}_i)^\top (\mathbf{y}_i - \mu - \mathrm{B}_i \mathbf{w}_i)] \tag{10}$$

$$= \log \mathcal{N}(\mathbf{y}_i \mid \mu + \mathrm{B}_i \mathrm{U}^\top \mathrm{K}_{pp}^{-1} \mathbf{k}_i, \sigma^2 \mathrm{I}_{n_i}) - \frac{\tilde{k}_{ii}}{2\sigma^2} \mathrm{Tr}[\mathrm{B}_i^\top \mathrm{B}_i] \tag{11}$$

$$\triangleq \log \tilde{p}(\mathbf{y}_i \mid \mathbf{u}, \mathbf{x}_i). \tag{12}$$

We can now write the lower bound on the conditional log-probability as

$$\log p(\mathbf{y} \mid \mathbf{u}, \mathbf{x}_{1:m}) \geq \sum_{i=1}^{m} \log \tilde{p}(\mathbf{y}_i \mid \mathbf{u}, \mathbf{x}_i) \triangleq \log \tilde{p}(\mathbf{y} \mid \mathbf{u}, \mathbf{x}_{1:m}). \tag{13}$$

To complete the derivation of the approximate objective, we use the lower bound on $\log p(\mathbf{y} \mid \mathbf{u}, \mathbf{x}_{1:m})$ to create a variational lower bound on the marginal log-probability of the trajectories

$$\log p(\mathbf{y}) = \log \int p(\mathbf{y} \mid \mathbf{u}, \mathbf{x}_{1:m}) p(\mathbf{u}, \mathbf{x}_{1:m}) d\mathbf{u} \tag{14}$$

$$\geq \int q(\mathbf{u}, \mathbf{x}_{1:m}) \left( \log p(\mathbf{y} \mid \mathbf{u}, \mathbf{x}_{1:m}) - \log q(\mathbf{u}, \mathbf{x}_{1:m}) + \log p(\mathbf{u}, \mathbf{x}_{1:m}) \right) d\mathbf{u} dx_{1:m} \tag{15}$$

$$\geq \int q(\mathbf{u}, \mathbf{x}_{1:m}) \left( \log \tilde{p}(\mathbf{y} \mid \mathbf{u}, \mathbf{x}_{1:m}) - \log q(\mathbf{u}, \mathbf{x}_{1:m}) + \log p(\mathbf{u}, \mathbf{x}_{1:m}) \right) d\mathbf{u} dx_{1:m} \tag{16}$$

$$\triangleq \log \tilde{p}(\mathbf{y}). \tag{17}$$

We assume that $\mathbf{u}, \mathbf{x}_1, \dots, \mathbf{x}_m$ are all mutually independent in the variational posterior. We use a multivariate normal variational approximation for each $\mathbf{x}_i$ with variational parameters $\mathbf{m}_i$ and $\mathrm{S}_i$.

Fixing $\mathbf{x}_i$, to find the the optimal form for $q(\mathbf{u})$, note that each $\log \tilde{p}(\mathbf{y}_i \mid \mathbf{u}, \mathbf{x}_i)$ is composed of a log-likelihood plus an additive term that is independent of $\mathbf{u}$. Therefore, the terms that depend on $\mathbf{u}$ can be written as:

$$\mathbb{E}_{q(\mathbf{u})} \left[ \sum_{i=1}^{m} \log \mathcal{N}(\mathbf{y}_i \mid \mu + \mathrm{B}_i \mathrm{U}^\top \mathrm{K}_{pp}^{-1} \mathbf{k}_i, \sigma^2 \mathrm{I}_{n_i}) \right] - \mathrm{KL}(q \| p). \tag{18}$$

Now, note that the mean in any of the log-likelihood terms can be rewritten as

$$\mu + \mathrm{B}_i \mathrm{U}^\top \mathrm{K}_{pp}^{-1} \mathbf{k}_i = \mu + (\mathrm{B}_i \otimes \mathbf{k}_i^\top \mathrm{K}_{pp}^{-1}) \mathbf{u}, \tag{19}$$

Let $\mathrm{C}_i \triangleq (\mathrm{B}_i \otimes \mathbf{k}_i^\top \mathrm{K}_{pp}^{-1})$ denote the *extended design matrix* obtained through this rewriting, and recall that each column $\mathbf{u}_k$ is normally distributed with mean zero and covariance $\mathrm{K}_{pp}$. The prior over the vectorized matrix $\mathbf{u}$ is therefore also multivariate normal. The expression above is maximized when $q(\mathbf{u})$ is equal to the posterior over $\mathbf{u}$ given the observed trajectories. Because the prior is multivariate normal and the mean of the likelihood depends linearly on $\mathbf{u}$, the posterior must also be multivariate normal. Moreover, we know its exact form:

$$\mathbf{m}_* = \mathrm{S}_* \left( \sigma^{-2} \sum_{i=1}^{m} \mathrm{C}_i^\top (\mathbf{y}_i - \mu) \right) \ , \ \mathrm{S}_* = \left( \sigma^{-2} \sum_{i=1}^{m} \mathrm{C}_i^\top \mathrm{C}_i + (\mathrm{I}_d \otimes \mathrm{K}_{pp}^{-1}) \right)^{-1}. \tag{20}$$

We therefore parameterize $q(\mathbf{u})$ as a multivariate normal distribution with variational parameters $\mathbf{m}$ and $\mathrm{S}$.

We now derive a closed-form expression for the expectation of $\log \tilde{p}(\mathbf{y}_i \mid \mathbf{u}, \mathbf{x}_i)$ under variational posterior distribution. Because $\mathbf{u}$ and $\mathbf{x}_i$ are assumed to be independent in the variational posteriors, we can analyze the expectation in either order. Fix $\mathbf{x}_i$, then we see that $\log \tilde{p}(\mathbf{y}_i \mid \mathbf{u}, \mathbf{x}_i)$ depends on $\mathbf{u}$ only through the mean of the Gaussian density, which is a quadratic term in log likelihood. Because $q(\mathbf{u})$ is multivariate normal, we can compute the expectation in closed form.

$$\mathbb{E}_{q(\mathbf{u})}[\log \tilde{p}(\mathbf{y}_i \mid \mathbf{u}, \mathbf{x}_i)] = \mathbb{E}_{q(\mathrm{U})}[\log \mathcal{N}(\mathbf{y}_i \mid \mu + (\mathrm{B}_i \otimes \mathbf{k}_i^\top \mathrm{K}_{pp}^{-1})\mathbf{u}, \sigma^2 \mathrm{I}_{n_i})] - \frac{\tilde{k}_{ii}}{2\sigma^2} \mathrm{Tr}[\mathrm{B}_i^\top \mathrm{B}_i]$$

$$= \log \mathcal{N}(\mathbf{y}_i \mid \mu + \mathrm{C}_i \mathbf{m}, \sigma^2 \mathrm{I}_{n_i})] - \frac{1}{2\sigma^2} \mathrm{Tr}[\mathrm{SC}_i^\top \mathrm{C}_i] - \frac{\tilde{k}_{ii}}{2\sigma^2} \mathrm{Tr}[\mathrm{B}_i^\top \mathrm{B}_i],$$

We can compute the expectation of $\mathbb{E}_{q(\mathbf{u})}[\log \tilde{p}(\mathbf{y}_i \mid \mathbf{u}, \mathbf{x}_i)]$ in closed form by noting that we need only compute expectations of $\mathbf{k}_i$ and $\mathbf{k}_i \mathbf{k}_i^\top$. Specifically, we have that

$$\mathbb{E}_{q(\mathbf{x}_i)}[k(\mathbf{x}_i, \mathbf{z}_j)] = \frac{\alpha}{|\mathrm{S}_i|^{1/2}|\mathrm{A}|^{1/2}} \exp\left\{\frac{1}{2}(\mathbf{B}^\top \mathrm{A}^{-1} \mathbf{b} - c)\right\}, \tag{21}$$

where $\mathrm{A} = \mathrm{S}_i^{-1} + \ell^{-2}\mathrm{I}_q$, $\mathbf{b} = \mathrm{S}_i^{-1}\mathbf{m}_i + \ell^{-2}\mathbf{z}_j$, and $c = \mathbf{m}_i^\top \mathrm{S}_i^{-1}\mathbf{m} + \ell^{-2}\mathbf{z}_j^\top \mathbf{z}_j$. Similarly, for the expected outer product, we have

$$\mathbb{E}_{q(\mathbf{x}_i)}[k(\mathbf{x}_i, \mathbf{z}_j)k(\mathbf{x}_i, \mathbf{z}_k)] = \frac{\alpha}{|\mathrm{S}_i|^{1/2}|\mathrm{A}|^{1/2}} \exp\left\{\frac{1}{2}(\mathbf{B}^\top \mathrm{A}^{-1} \mathbf{b} - c)\right\}, \tag{22}$$

where $\mathrm{A} = \mathrm{S}_i^{-1} + 2\ell^{-2}\mathrm{I}_q$, $\mathbf{b} = \mathrm{S}_i^{-1}\mathbf{m}_i + \ell^{-2}\mathbf{z}_j + \ell^{-2}\mathbf{z}_k$, and $c = \mathbf{m}_i^\top \mathrm{S}_i^{-1}\mathbf{m} + \ell^{-2}\mathbf{z}_j^\top \mathbf{z}_j + \ell^{-2}\mathbf{z}_k^\top \mathbf{z}_k$. Importantly, we can simply substitute these expectations into $\mathbb{E}_{q(\mathbf{u})}[\log \tilde{p}(\mathbf{y}_i \mid \mathbf{u}, \mathbf{x}_i)]$ and the form of the lower bound does not change (it is still a Gaussian log-likelihood plus the additional trace terms).

## 2  Optimizing the Evidence Lower Bound

To formulate the complete objective, we use the lower bound derived above and place priors on the observation noise $\sigma^2$, and the hyperparameters of the kernel $k(\cdot, \cdot)$. In this section and in our experiments we assume that the kernel is a radial basis function (RBF) with scale $\alpha$ and length-scale (or bandwidth) $\ell$. We assume normal distributions over the log of $\sigma^2$, $\alpha$, and $\ell$ with mean parameters $m_s$, $m_a$, $m_\ell$ respectively and precision parameters $\rho_s$, $\rho_a$, and $\rho_\ell$ respectively. Our objective is therefore

$$\mathcal{J}_{\text{SA-DTM}}(\mathbf{m}, \mathrm{S}, \mathbf{m}_{1:m}, \mathrm{S}_{1:m}, \mu, \sigma^2, \alpha, \ell) = \tag{23}$$

$$\sum_{i=1}^m -\frac{n_i}{2}\log 2\pi\sigma^2 - \frac{1}{2\sigma^2}\mathbb{E}_{q(\mathbf{x}_i)}[\|\mathbf{y}_i - \mu - (\mathrm{B}_i \otimes \mathbf{k}_i^\top \mathrm{K}_{pp}^{-1})\mathbf{m}\|_2^2] \tag{24}$$

$$+ \sum_{i=1}^m -\frac{1}{2\sigma^2}\mathrm{Tr}[\mathrm{S}(\mathrm{B}_i^\top \mathrm{B}_i \otimes \mathrm{K}_{pp}^{-1}\mathbb{E}_{q(\mathbf{x}_i)}[\mathbf{k}_i \mathbf{k}_i^\top]\mathrm{K}_{pp}^{-1})] \tag{25}$$

$$+ \sum_{i=1}^m -\frac{1}{2\sigma^2}\mathrm{Tr}[\mathrm{B}_i^\top \mathrm{B}_i](\alpha - \mathbb{E}_{q(\mathbf{x}_i)}[\mathbf{k}_i^\top \mathrm{K}_{pp}^{-1}\mathbf{k}_i]) \tag{26}$$

$$- \sum_{i=1}^m \frac{1}{2}\left(\mathrm{Tr}[\mathrm{S}_i + \mathbf{m}_i \mathbf{m}_i^\top] - q - \log|\mathrm{S}_i|\right) \tag{27}$$

$$- \frac{1}{2}\left(\mathrm{Tr}[(\mathrm{S} + \mathbf{m}\mathbf{m}^\top)(\mathrm{I}_d \otimes \mathrm{K}_{pp}^{-1})] - pd + \log\frac{|\mathrm{K}_{pp}|^d}{|\mathrm{S}|}\right) \tag{28}$$

$$- \frac{\rho_s}{2}\|\log\sigma^2 - m_s\|_2^2 - \frac{\rho_a}{2}\|\log\alpha - m_a\|_2^2 - \frac{\rho_\ell}{2}\|\log\ell - m_\ell\|_2^2. \tag{29}$$

Note that the last three lines above can be seen as regularizers (log priors for the hyperparameters and a KL divergence between the variational distribution $q$ and the prior $p$). The first four lines can be decomposed across individuals, suggesting that we can use stochastic approximation of the objective and its gradients to derive a scalable algorithm for optimizing the objective.

We define an iterative first-order optimization algorithm. In broad strokes, within each iteration we will sample a single individual $i$ (or a batch of patients), maximize the objective with respect to $\mathbf{m}_i$

and $S_i$ while holding the global variables fixed, compute the approximate gradients of the objective, and take a small step in the direction of each gradient for each parameter (the step size is determined by a learning schedule, which may be specific to each global variable). We discuss each step in detail below. We do so assuming a single sampled individual $i$, although in principle we can sample a batch of individuals to reduce variance in the gradient estimate.

**Maximizing wrt local variables $(\mathbf{m}_i, S_i)$.** Before computing gradients of the approximate objective with respect to the global parameters, we first do a block coordinate optimization over the local variational parameters of individual $i$. We optimize:

$$J_i(\mathbf{m}_i, S_i) = \tag{30}$$

$$-\frac{n_i}{2} \log 2\pi\sigma^2 - \frac{1}{2\sigma^2} \mathbb{E}_{q(\mathbf{x_i})}[\|\mathbf{y}_i - \mu - (B_i \otimes \mathbf{k}_i^\top K_{pp}^{-1})\mathbf{m}\|_2^2] \tag{31}$$

$$-\frac{1}{2\sigma^2} \operatorname{Tr}[S(B_i^\top B_i \otimes K_{pp}^{-1}\mathbb{E}_{q(\mathbf{x_i})}[\mathbf{k}_i\mathbf{k}_i^\top]K_{pp}^{-1})] \tag{32}$$

$$-\frac{1}{2\sigma^2} \operatorname{Tr}[B_i^\top B_i](\alpha - \mathbb{E}_{q(\mathbf{x_i})}[\mathbf{k}_i^\top K_{pp}^{-1}\mathbf{k}_i]). \tag{33}$$

We can optimize this expression using a gradient-based optimizer. We use the scaled conjugate gradients algorithm.

**Estimating gradients of global variables.** Having sampled individual $i$ and having refit her local variational parameters, we now want to approximate the gradient of the full objective with respect to the global variables $\mathbf{m}$, $S$, $\mu$, $\sigma^2$, $\alpha$, and $\ell$. We first look at the approximate gradient with respect to $\mathbf{m}$.

$$\hat{\nabla}_{\mathcal{J}_{\text{SA-DTM}}}(\mathbf{m}) = \mathbb{E}_{q(\mathbf{x}_i)}[\frac{m}{\sigma^2}(B_i^\top \otimes K_{pp}^{-1}\mathbf{k}_i)(\mathbf{y}_i - \mu - (B \otimes \mathbf{k}_i^\top K_{pp}^{-1})\mathbf{m})] - (I_d \otimes K_{pp}^{-1})\mathbf{m}. \tag{34}$$

The approximate gradient with respect to $S$ is

$$\hat{\nabla}_{\mathcal{J}_{\text{SA-DTM}}}(S) = -\frac{m}{2\sigma^2} \operatorname{Tr}[(B_i^\top B_i \otimes K_{pp}^{-1}\mathbb{E}_{q(\mathbf{x}_i)}[\mathbf{k}_i\mathbf{k}_i^\top]K_{pp}^{-1})] \tag{35}$$

$$-\frac{1}{2} \operatorname{Tr}[(I_d \otimes K_{pp}^{-1})] + \frac{1}{2} \operatorname{Tr}[S^{-1}]. \tag{36}$$

Note that if we set these approximate gradients to $0$, we obtain the following estimates of $\mathbf{m}$ and $S$:

$$\hat{\mathbf{m}} = \hat{S}\left(\frac{m}{\sigma^2}(B_i^\top \otimes K_{pp}^{-1}\mathbb{E}_{q(\mathbf{x}_i)}[\mathbf{k}_i])(\mathbf{y} - \mu)\right) \tag{37}$$

$$\hat{S} = \left(\frac{m}{\sigma^2}(B_i^\top B_i \otimes K_{pp}^{-1}\mathbb{E}_{q(\mathbf{x}_i)}[\mathbf{k}_i\mathbf{k}_i^\top]K_{pp}^{-1}) + (I_d \otimes K_{pp}^{-1})\right)^{-1} \tag{38}$$

We can improve the rate of convergence of our algorithm by taking the geometry of the space of distributions parameterized by $\mathbf{m}$ and $S$ into account. We do so by using the *natural gradients* for these two parameters instead of the approximations above. Let $\boldsymbol{\theta}_1$ and $\theta_2$ denote the canonical parameterization of the variational multivariate normal, then the gradient updates at time $t$ are Hoffman et al. [2013]:

$$\boldsymbol{\theta}_1^t = \boldsymbol{\theta}_1^{t-1} + \lambda_t(\eta_1^{t-1} - \boldsymbol{\theta}_1^{t-1}) \tag{39}$$

$$\theta_2^t = \theta_2^{t-1} + \lambda_t(\eta_2^{t-1} - \theta_2^{t-1}), \tag{40}$$

where

$$\eta_1^{t-1} = \frac{m}{\sigma^2}(B_i^\top \otimes K_{pp}^{-1}\mathbb{E}_{q(\mathbf{x}_i)}[\mathbf{k}_i])(\mathbf{y} - \mu) \tag{41}$$

$$\eta_2^{t-1} = -\frac{m}{2\sigma^2}(B_i^\top B_i \otimes K_{pp}^{-1}\mathbb{E}_{q(\mathbf{x}_i)}[\mathbf{k}_i\mathbf{k}_i^\top]K_{pp}^{-1}) \tag{42}$$

To update the hyperparamters, we need to compute the gradients with respect to $\mu$, $\sigma^2$, $\alpha$, and $\ell$. We parameterize $\sigma^2$, $\alpha$, and $\ell$ using their logarithms, and so present gradients with respect to that representation. To make the expressions more clear, we present the gradients as differentials with

respect to the kernel, which can be completed using the chain rule. The estimate of the gradient with respect to $\mu$ is

$$\hat{\nabla}_{\mathcal{J}_{\text{SA-DTM}}}(\mu) = \frac{m}{\sigma^2}(\mathbf{y}_i - \mu - (\mathrm{B}_i \otimes \mathbb{E}_{q(\mathbf{x}_i)}[\mathbf{k}_i^\top]\mathrm{K}_{pp}^{-1})\mathbf{m})^\top \mathbf{1}_{n_i}. \tag{43}$$

The estimate of the gradient with respect to $\log \sigma^2$ is

$$\hat{\nabla}_{\mathcal{J}_{\text{SA-DTM}}}(\log \sigma^2) = -\frac{mn_i}{2} + \frac{m}{2\sigma^2}\mathbb{E}_{q(\mathbf{x}_i)}[\|\mathbf{y}_i - \mu - (\mathrm{B}_i \otimes \mathbf{k}_i^\top \mathrm{K}_{pp}^{-1})\mathbf{m}\|_2^2] \tag{44}$$

$$+ \frac{m}{2\sigma^2}\operatorname{Tr}[\mathrm{S}(\mathrm{B}_i^\top \mathrm{B}_i \otimes \mathrm{K}_{pp}^{-1}\mathbb{E}_{q(\mathbf{x}_i)}[\mathbf{k}_i\mathbf{x}_i^\top]\mathrm{K}_{pp}^{-1})] \tag{45}$$

$$+ \frac{m}{2\sigma^2}\operatorname{Tr}[\mathrm{B}_i^\top \mathrm{B}](\alpha - \operatorname{Tr}[\mathrm{K}_{pp}^{-1}\mathbb{E}_{q(\mathbf{x}_i)}[\mathbf{k}_i\mathbf{x}_i^\top]]) \tag{46}$$

$$- \rho_s(\log \sigma^2 - m_s). \tag{47}$$

The estimate of the gradient with respect to $\log \alpha$ is

$$\hat{\nabla}_{\text{SA-DTM}}(\log \alpha) = \tag{48}$$

$$\frac{m}{\sigma^2}\mathbb{E}_{q(\mathbf{x}_i)}[(\mathbf{y}_i - \mu - \mathrm{Cm})^\top(\mathrm{B}_i \otimes \partial\mathbf{k}_i^\top \mathrm{K}_{pp}^{-1} - \mathbf{k}_i^\top \mathrm{K}_{pp}^{-1}\partial\mathrm{K}_{pp}\mathrm{K}_{pp}^{-1})\mathbf{m}] \tag{49}$$

$$- \frac{m}{\sigma^2}\mathbb{E}_{q(\mathbf{x}_i)}[\operatorname{Tr}[\mathrm{SC}_i^\top(\mathrm{B}_i \otimes \partial\mathbf{k}_i^\top \mathrm{K}_{pp}^{-1} - \mathbf{k}_i^\top \mathrm{K}_{pp}^{-1}\partial\mathrm{K}_{pp}\mathrm{K}_{pp}^{-1})]] \tag{50}$$

$$- \frac{m}{2\sigma^2}\operatorname{Tr}[\mathrm{B}_i^\top \mathrm{B}_i]\alpha \tag{51}$$

$$+ \frac{m}{2\sigma^2}\operatorname{Tr}[\mathrm{B}_i^\top \mathrm{B}_i](2\mathbb{E}_{q(\mathbf{x}_i)}[\mathbf{k}_i^\top]\mathrm{K}_{pp}^{-1}\partial\mathbb{E}_{q(\mathbf{x}_i)}[\mathbf{k}_i] - \operatorname{Tr}[\mathrm{K}_{pp}^{-1}\partial\mathrm{K}_{pp}\mathrm{K}_{pp}^{-1}\mathbb{E}_{q(\mathbf{x}_i)}[\mathbf{k}_i\mathbf{x}_i^\top]]) \tag{52}$$

$$+ \frac{1}{2}\left(\operatorname{Tr}[(\mathrm{S} + \mathrm{mm}^\top)(\mathrm{I}_d \otimes \mathrm{K}_{pp}^{-1}\partial\mathrm{K}_{pp}\mathrm{K}_{pp}^{-1})] - d\operatorname{Tr}[\mathrm{K}_{pp}^{-1}\partial\mathrm{K}_{pp}]\right). \tag{53}$$

The estimate of the gradient with respect to $\log \ell$ is

$$\hat{\nabla}_{\text{SA-DTM}}(\log \ell) = \tag{54}$$

$$\frac{m}{\sigma^2}\mathbb{E}_{q(\mathbf{x}_i)}[(\mathbf{y}_i - \mu - \mathrm{Cm})^\top(\mathrm{B}_i \otimes \partial\mathbf{k}_i^\top \mathrm{K}_{pp}^{-1} - \mathbf{k}_i^\top \mathrm{K}_{pp}^{-1}\partial\mathrm{K}_{pp}\mathrm{K}_{pp}^{-1})\mathbf{m}] \tag{55}$$

$$- \frac{m}{\sigma^2}\mathbb{E}_{q(\mathbf{x}_i)}[\operatorname{Tr}[\mathrm{SC}_i^\top(\mathrm{B}_i \otimes \partial\mathbf{k}_i^\top \mathrm{K}_{pp}^{-1} - \mathbf{k}_i^\top \mathrm{K}_{pp}^{-1}\partial\mathrm{K}_{pp}\mathrm{K}_{pp}^{-1})]] \tag{56}$$

$$+ \frac{m}{2\sigma^2}\operatorname{Tr}[\mathrm{B}_i^\top \mathrm{B}_i](2\mathbb{E}_{q(\mathbf{x}_i)}[\mathbf{k}_i^\top]\mathrm{K}_{pp}^{-1}\partial\mathbb{E}_{q(\mathbf{x}_i)}[\mathbf{k}_i] - \operatorname{Tr}[\mathrm{K}_{pp}^{-1}\partial\mathrm{K}_{pp}\mathrm{K}_{pp}^{-1}\mathbb{E}_{q(\mathbf{x}_i)}[\mathbf{k}_i\mathbf{x}_i^\top]]) \tag{57}$$

$$+ \frac{1}{2}\left(\operatorname{Tr}[(\mathrm{S} + \mathrm{mm}^\top)(\mathrm{I}_d \otimes \mathrm{K}_{pp}^{-1}\partial\mathrm{K}_{pp}\mathrm{K}_{pp}^{-1})] - d\operatorname{Tr}[\mathrm{K}_{pp}^{-1}\partial\mathrm{K}_{pp}]\right). \tag{58}$$