[Reviews · NeurIPS 2016]

Reviewer 1

Summary

In short, the main contribution of the paper is a framework for learning a low-dimensional representation of time series data. In this work, the authors propose a probabilistic model that learns a low-dimensional representation of univariate time-series data. The authors learn the model using a stochastic variational inference algorithm tailored to their application, which allows the model to scale to large datasets. Applied to a real world dataset of patients with a complex autoimmune disease, the model appears to learn clinically meaningful representations of disease trajectories.

Qualitative Assessment

Overall, this is an important and interesting problem. Increasingly, researchers in ML and healthcare are studying methods for learning from longitudinal (time series) data (as opposed to more traditional approaches that consider only snapshot data). The work is well motivated, and clearly presented. Moreover, the authors include experiments on real data pertaining to patients with a complex chronic disease. Evaluation is two-fold. First look at clusters obtained by clustering low-dimensional representation, and then test relationship between low-dimensional representation and outcome. However, I do have a few concerns regarding this evaluation: *The authors state that the median number of observations is 2 for TSS data and 3 for PFVC data. This is somewhat concerning given that the proposed model is primarily meant to model trajectories. If half of the patients have only two or fewer measurements, is this really a trajectory? Have the authors experimented with other datasets, or perhaps simulated data? *Another related concern, is whether it makes sense clinically to compare patients who have 2 measurements with patients who have 22 measurements? Are these even clinically comparable? How does the number of measurements vary across clusters? *In this particular example, it seems like we could characterize PFVC and TSS trajectories based on extent of changes in measurements and when they occur. Do we really need DTMs to help us identify declining trajectories vs. recovering trajectories? Again, I can see that while these trajectories are relatively simple, there may be disease processes with more complex trajectories to which the proposed method could be applied. Simulations may help show this. *Quantitative evaluation only sheds so much light on the ability of the low dimensional representation to identify important clinical subgroups. In order to quantitatively evaluate clusters, why not look at outcome purity within each cluster? This idea is somewhat explored in the last figure of the paper, but could be interesting to show the frequency of outcomes within each of the clusters in Figures 1 & 2. *In the introduction DTM is described as a method for modeling the trajectory of a single clinical marker. How do the clusters captured by PFVC and TSS compare? Can the proposed method capture trends in multivariate trajectories?

Confidence in this Review

2-Confident (read it all; understood it all reasonably well)


Reviewer 2

Summary

The manuscript, "Disease Trajectory Maps", describes the authors' work to develop a low rank representation for irregularly sampled time series of disease state using a novel extension of the GPLVM method to linear mixed models. Clustering in the latent representation space allows for identification of similar disease trajectories, which are demonstrated to be predictive of critical end points in an application to scleroderma.

Qualitative Assessment

The development of fast, sophisticated statistical methods for understanding irregular time series of noisy observations is an important problem for applied statistics in the healthcare sector. Previous studies have explored a wide variety of methods which have tended to trade-off statistical sophistication for model flexibility; whereas the present effort pushes forwards on both fronts, combing the power of hierarchical Bayesian modelling over a B-spline basis with contemporary methods for low rank representations, 'GP sparsity' (inducing points), and variational inference. I expect this method will be of interest both within the machine learning community and at its intersection with the healthcare/epidemiology world. One slight concern I have is that a non-trivial body of further work seems to remain to continue model development towards practical, clinically useable results; for instance, the identification of clusters in the latent space is (by my understanding) at present simply by a 'post hoc' algorithmic method with the known end points reserved only for performance testing. Likewise, from the clinical persepective one must be interested in the accurate calibration of posterior uncertainties with respect to cluster type and probable trajectory. Minor notes: - The sentence on line 155 ("This reformulation ... suggests a natural alternative to learning the representations x_i ....") seems confusing since the model does learn the representations, x_i? Do you mean simply that it's an alternative to LMM? - Some comment to share your insights on the possible impact of the variational approximation on this particular model would be welcome.

Confidence in this Review

2-Confident (read it all; understood it all reasonably well)


Reviewer 3

Summary

The paper presents the Disease Trajectory Map (DTM) which models the infrequently and irregularly sampled time series data generated by clinical markers. These clinical markers characterise the progression of medical conditions. The goal of the DTM is the generation of visualisations that can assist clinicians in tracking disease progression. This would be of particular benefit where available markers do not on their own necessarily well predict progression. The DTM reformulates a linear mixed model that assumes Gaussian measurement noise and employs the reduced rank model for dimensionality reduction. To ensure scalability the authors use a stochastic variational inference algorithm, and approximate the posterior distribution, followed by Monte Carlo estimation to marginalise the predicted trajectories.

Qualitative Assessment

The paper is solid, and I would recommend it for publication. The model is most similar to the Gaussian process latent variable model but the authors point out the novelty is the adaptation of the approach to functional time series data. The application is also a valuable one, and the model has discovered clinically verified relationships between certain biomarkers in predicting the course of scleroderma that were not discovered by other models.

Confidence in this Review

2-Confident (read it all; understood it all reasonably well)


Reviewer 4

Summary

The authors model disease trajectories in a disease trajectory map. The model is built using GPLVM. The resulting map appears to be a useful tool for disease analysis.

Qualitative Assessment

The authors apply the term trajectory in a completely unusual way. The reader may confuse it with spatial trajectories, it is important to highlight this difference. A map usually has the advantage that future trajectories can be located easily within this map, The authors could analyse how their resulting map behaves on incremental updates. In addition it is interesting to identify decision points and vertices among these trajectories. The usage of their resulting maps could be discussed in more detail. Already the introduction could be improved by better motivation of their research question. Why is it useful to have these maps? Who uses them? what for?

Confidence in this Review

2-Confident (read it all; understood it all reasonably well)


Reviewer 5

Summary

This paper aims to learn an embedding such that trajectories can be compared. They start by expressing each trajectory as a linear mixed model. They then show how by marginalization, the joint distribution over trajectories is a Gaussian process. By using recent methods for scaling Gaussian processes (inducing points) along with stochastic variational inference, they learn a variational distribution over both the inducing points and the embeddings.

Qualitative Assessment

The problem is important from a medical standpoint: it relates to phenotyping patients, which is an important area in computational medicine. The method seems reasonable and isn't obvious, and the results seem decent, although as I'm not a medical expert, some of the interpretation is difficult for me. My main gripe is that the clarity isn't great. As it stands, the summary I provided (assuming it's correct) wasn't easy to tease out from the way the paper is currently written: such a summary would be good under your 'contributions' section. For the figures, other than the trajectories vs. the red lines which are clear, it's difficult to tell what good results would look like. Finally, there is too much of the paper spent on background and related work: at least some of that could be spent on explaining linear mixed models (i.e. the equation form rather than only the distribution form) and inducing points more clearly.

Confidence in this Review

2-Confident (read it all; understood it all reasonably well)


Reviewer 6

Summary

This paper looks at the problem of trajectory clustering. It first introduce a number of prexisting model. And introduces the DTM model.

Qualitative Assessment

There's a lot of part of this article that I find hard to understand. This paper lacks clarity. What the author is looking to optimise is unclear.(I think it's equation 5 but since this is not stated explicitely I'm not sure) The motivation for using this new model is not lear either. This should appear more explicitely. The way this article is structured it has a lot of big blocks of text with not paragraph this doesn't help comprehension.

Confidence in this Review

1-Less confident (might not have understood significant parts)